# Peroxygenase-Catalyzed Selective Synthesis of Calcitriol Starting from Alfacalcidol

**DOI:** 10.3390/antiox11061044

**Published:** 2022-05-25

**Authors:** Yuanying Li, Pengpeng Zhang, Zhoutong Sun, Huanhuan Li, Ran Ge, Xiang Sheng, Wuyuan Zhang

**Affiliations:** 1Tianjin Institute of Industrial Biotechnology, Chinese Academy of Sciences, Tianjin 300308, China; liyy@tib.cas.cn (Y.L.); zhangpp@tib.cas.cn (P.Z.); sunzht@tib.cas.cn (Z.S.); li-huan@stu.xjtu.edu.cn (H.L.); ger@tib.cas.cn (R.G.); shengx@tib.cas.cn (X.S.); 2National Center of Technology Innovation for Synthetic Biology, Tianjin 300308, China; 3School of Chemical Engineering and Technology, Xi’an Jiaotong University, Xi’an 710000, China

**Keywords:** peroxygenases, oxyfunctionalization, active vitamin D, calcitriol, biocatalysis

## Abstract

Calcitriol is an active analog of vitamin D3 and has excellent physiological activities in regulating healthy immune function. To synthesize the calcitriol compound, the concept of total synthesis is often adopted, which typically involves multiple steps and results in an overall low yield. Herein, we envisioned an enzymatic approach for the synthesis of calcitriol. Peroxygenase from *Agrocybe aegerita* (*Aae*UPO) was used as a catalyst to hydroxylate the C-H bond at the C-25 position of alfacalcidol and yielded the calcitriol in a single step. The enzymatic reaction yielded 80.3% product formation in excellent selectivity, with a turnover number up to 4000. In a semi-preparative scale synthesis, 72% isolated yield was obtained. It was also found that *Aae*UPO is capable of hydroxylating the C-H bond at the C-1 position of vitamin D3, thereby enabling the calcitriol synthesis directly from vitamin D3.

## 1. Introduction

Vitamin D, also known as the “sunshine vitamin”, is well-known for its pivotal role in healthy immune function [1]. In addition to promoting bone health in children and adults, it provides other health benefits, including reducing the risk of chronic diseases such as cancer, autoimmune, or cardiovascular diseases [2,3,4]. Though vitamin D is essential to human health and well-being, the deficiency of vitamin D is a widespread health problem [5,6]. On the other hand, in comparison to conventional vitamin D, the hydroxylated vitamin Ds (also known as active vitamin Ds), which are metabolites after the intake of vitamin Ds, has significantly stronger physiological activity [7]. For example, based on the structure of natural vitamin D3, the selective hydroxylation at C-25, C-1, or both sites yields 25-hydroxyvitamin D3, alfacalcidol, and calcitriol, respectively (Figure 1). The positions accessible to hydroxylation reactions thus lead to various forms of active vitamin D3s. All active vitamin D3s are mainly used clinically for the treatment of chronic renal insufficiency, parathyroidism, rickets, chondromalacia, and bone lesions caused by abnormal vitamin D3 metabolism. Calcitriol, for example, is 500–1000 times more active in calcium absorption, compared with 25-hydroxyvitamin D3 (Figure 1) [8,9]. In addition, calcitriol is also used as an antitumor agent, fat emulsion, sebum secretion promoter, etc. [10].

The abovementioned hydroxylation of vitamin D3, in essence, is the selective C-H bond activation and oxygenation, which is one of the most challenging reactions in synthetic chemistry. To synthesize calcitriol, the chemical methods often rely on the concept of total synthesis to obtain the final molecular skeleton by the convergent synthesis method [11,12]. Since the total synthesis method requires separate synthesis of A-ring and C/D-ring, the synthetic route is typically tedious and results in an overall low yield.

In nature, a broad scope of enzymes is available to catalyze selective oxyfunctionalization of C-H bonds [13,14,15]. The chemo-, regio- and stereo-selectivity make enzymes ideal catalysts in various organic syntheses. Among the reported enzymes, P450 monooxygenases capable of selective hydroxylation of the VD3 at C-1 and C-25 positions have been identified in natural species such as bacteria, yeast, and mammals [16,17,18]. However, the need for a specific electron transport chain involving NAD(P)H cofactors in the catalytic cycle of P450 enzymes leaves limited room for process optimization. Thus far, the potential for industrial application of P450s still remains unexplored [19]. In contrast, unspecific peroxygenases (UPOs) have been considered “ideal” biocatalysts for oxidative functionalization reactions of a variety of activated and nonactivated C-H bonds [20,21,22,23,24,25,26]. A recent report by Hollmann et al. substantially demonstrated the potential of large-scale production of the enzyme yielding practical amounts of these promising catalysts [27]. UPOs share the same active site characteristics (e.g., cysteine sulfur atoms as heme iron ligands) and common reaction chemistry as P450 monooxygenases [28]. UPOs have the unique advantage of only relying on peroxides (H_2_O_2_, ROOH) to introduce an oxygen atom to different target molecules; therefore, the reaction scheme is significantly simplified, compared with P450s [29]. Pleasantly, peroxygenases from *Agrocybe aegerita* (*Aae*UPO) and *Coprinopsis cinerea* (*Cci*UPO) have shown to be able to accept vitamin D3 as substrate and yield 25-hydroxyvitamin D3, in which *Cci*UPO excels in selectivity and catalytic activity [30,31]. However, the potential of using peroxygenases for the synthesis of active D3 compounds remains significantly underdeveloped.

With a research interest in the challenging chemistry of the synthesis of active vitamin D3s, as well as the need for a straightforward synthetic route avoiding protection/deprotection procedures, herein we aimed at exploring the use of peroxygenase in the synthesis of calcitriol from alfacalcidol (Figure 1, pathway I). Ideally, the synthesis could also start from vitamin D3 as substrate via 25-hydroxyvitamin D3 using the same enzyme and two equivalents of cosubstrate H_2_O_2_ (Figure 1, in combination with pathways II and I). Thus, aside from new substrates (alfacalcidol) investigated for peroxygenase, new synthetic pathways are also demonstrated for the synthesis of active vitamin D3 compounds (calcitriol).

## 2. Materials and Methods

### 2.1. Materials

In all experiments, the vitamin D3, alfacalcidol, 25-hydroxyvitamin D3, and calcitriol were purchased from Aladdin. Other reactants and solvents were obtained from commercial sources.

### 2.2. Production and Purification of AaeUPO

The unspecific peroxygenase from *Agrocybe aegerita* (*Aae*UPO, main isoform II) was produced and purified according to the procedures described previously [32]. After fermentation, a culture broth with *Pichia pastoris* cells containing *Aae*UPO was separated via centrifugation (8000 rpm, 2 h, 4 °C). Afterward, the supernatant was filtered through a 22 µm filter and kept at −80 °C. *Aae*UPO activity was determined to be 1106 ± 27 U mg^−1^ (pH 5.0 in NaPi buffer). (Sigma-Aldrich, Darmstadt, Germany). One unit of the enzyme activity was defined as the amount of the enzyme catalyzing the oxidation of 1 µmol of ABTS per minute (25 °C, pH 5.0). To purify the enzyme, the supernatant was first concentrated (Sartorius, Göttingen, Germany, 10 kDa cutoff) and dialyzed against 100 mM sodium phosphate, pH 5.0 *Aae*UPO was purified using an NGC Chromatography system (Bio-Rad, Hercules, CA, USA). The separation was performed on a Q Sepharose FF 30-mL cartridge, with a flow rate of 5 mL min^−1^. After 90 mL, the retained protein was eluted with a 0–50% NaCl gradient in 450 mL, followed by a 50–100% gradient in 50 mL and 100% NaCl in 75 mL. The purified UPO was confirmed by sodium dodecyl sulfate (SDS)–PAGE and stained with Coomassie brilliant blue R-250 (Sigma-Aldrich, Darmstadt, Germany).

### 2.3. Enzymatic Calcitriol Synthesis from Alfacalcidol and Vitamin D3

Alfacalcidol was dissolved in 1mL NaPi buffer (50 mM, pH 6) containing 40% (*v*/*v*) acetone. Next, *Aae*UPO was added to the above mixture, and hydrogen peroxide solution was supplied to the reaction using a syringe pump at a rate of 1.0 mM h^−1^. The reaction mixture was mixed in a thermal shaker at 25 °C, 800 rpm. The final reaction condition was: alfacalcidol (2.5 mM), H_2_O_2_ (5 mM), 40% acetone, NaPi pH 6 (100 mM), 25 °C. At different intervals (1, 3, 5, 8, and 24 h), 50 μL of the reaction mixture was withdrawn and extracted with 100 μL of ethyl acetate. The organic phase was dried over Na_2_SO_4_ and then evaporated completely. Then, the compound was redissolved in 200 μL of ethanol and subjected to HPLC analysis. The product concentration of calcitriol was determined via liquid chromatography. A similar approach was used when vitamin D3 was used as the substrate. The calcitriol concentration of the reactions was determined by a calibration curve. The formation was calculated by [product]/([substrate] + [product]) × 100%.

For a semi-preparative scale synthesis, 200 mg of alfacalcidol was dissolved in 200 mL of NaPi buffer (pH 7, 100 mM) containing 40% acetone. H_2_O_2_ was dosed at a rate of 1.5 mM h^−1^ at 25 °C for 8 h under magnetic stirring. Then, the reaction mixture was extracted with 200 mL of dichloromethane two times. The organic phase was concentrated, and the obtained oily mixture was purified by silica column (petroleum ether:ethyl acetate = 1:1). Thus, 150 mg of the final product calcitriol was obtained as a white powder.

### 2.4. Enzyme Activity Assays

The absorbance value was measured at 25 °C and 420 nm using an Infinite M200 Pro microplate reader. Briefly, 500 nM of peroxygenase was added to a transparent polypropylene 96-well screening plate, and 202 μL of screening solution (final: 100 mM citric acid-phosphate buffer pH 4.4; 1 mM azino-bis(3-ethylbenzothiazoline-6-sulfonic acid (ABTS); 1 mM H_2_O_2_) was added. Absorption values at420 nm of each well were immediately measured after enzyme addition under a kinetic mode (measurement interval: 30 s) over a duration of 5 to 12 min using the microtiter plate reader. The slope value corresponding to the increase in the absorption was obtained, and enzyme activity was calculated using the enzyme activity formula (Enzyme activity = E_w_ × V_total_ × V_sample_^−1^ × ε^−1^ × *d*^−1^, where V_sample_ is the volume of enzyme of the reaction solution to be measured; ε is the extinction coefficient of ABTS at 420 nm (ε = 36,000 M^−1^ × cm^−1^); d is the cuvette diameter (*d* = 1 cm); V_total_ is the total volume of the assay system(V_total_ = 202 μL); E_w_ is the sample absorbance).

### 2.5. HPLC Analysis

The high-performance liquid chromatography (HPLC) analysis was carried out on LC-2030 Plus Shimadzu equipped with a Shim-pack GIST C18 column (250 mm × 4.6 mm, 5 μm). The substrate and products were measured at 265 nm on LC-2030 Plus equipped with a UV detector (190 to 700 nm). The flow rate of the mobile phase was 1 mL min^−1^, and the column temperature was kept at 35 °C. The injection volume was 10 μL. The sample was dissolved in 200 μL ethanol. Water and acetonitrile were used as solvents A and B, respectively. The gradient program was as follows: 45% A and 55% B from 0 to 25 min; 15% A and 85% B from 45 to 60 min; and 45% A and 55% B from 65 to 80 min. Alfacalcidol and calcitriol were used as standard compounds to perform standard calibration curves for quantification. It should be noted that a particular challenge emerged regarding vitamin D3 data collection in the HPLC analysis. The retention time of vitamin D3 was longer than 100 min with shifting replicability; therefore, to obtain reliable data, when the calcitriol was synthesized from vitamin D3, the product formation was based on the calibration curves of 25-hydroxyvitamin D3 and calcitriol.

### 2.6. LC–MS Analysis

LC–MS/MS analysis was carried out using a Bruker Metabolic Profiler (Agilent 1260-Sprak Prospekt2-Bruker AVANCE III 600MHz/Bruker micrOTOF-Q II) (Agilent, Santa Clara, CA, USA). The obtained product was dissolved in ethanol. Chromatographic separation was performed using a Shim-pack GIST C18 column (250 mm × 4.6 mm, 5 μm) maintained at 35 °C and a G1315D UV–VIS detector at 265 nm. The injection volume was 10 μL. The mobile phase was ultrapure water (A)/acetonitrile (B), and the flow rate was 1 mL/min. The gradient program was as follows: 45% A and 55% B from 0 to 25 min; 15% A and 85% B from 45 to 60 min; 45% A and 55% B from 65 to 80 min. The mass spectrometer was operated in positive ion mode using electrospray ionization (ESI). The parameters for mass detection in positive ion mode were as follows: drying gas flow rate, 6 L/min; detector voltage, −4.0 kV; dry temperature, 180 °C. For MS analysis, collision energy of 12 V was applied. The multiple reaction monitoring (MRM) mode was used to detect the sample. LabSolutions Insight Version 2.0 software was used for the quantitative analysis of the data.

### 2.7. Docking

Molecular docking was performed on the basis of the crystal structure of Pada-I (PDB ID: 5OXU) [33]. The grid map with 60 × 60 × 60 points and a grid-point spacing of 0.375 Å was centered at the Fe atom by AutoGrid 4.2 [34]. Then, AutoDock 4.2 [34] was used for the docking simulations. The best-scored docking pose was acquired from each docking for the analysis.

## 3. Results and Discussion

In the first set of experiments, we recombinantly produced the Pada-I variant of peroxygenase from *Pichia pastoris* (*Aae*UPO) [32] and used alfacalcidol (**2**) as a model substrate to synthesize calcitriol (**4**). Though a number of methods for in situ provision of H_2_O_2_ have been demonstrated to sustain peroxygenase [35,36,37,38,39,40,41,42], we chose a direct addition of H_2_O_2_ via a syringe pump to simplify the reaction system. Under an arbitrarily chosen reaction condition (2.5 mM loading of **2**), the formation of calcitriol was continuously observed over the time course as monitored by reserved phase HPLC (Figure 2A). Overall, 80.3% product formation (2.0 mM) was achieved in 8 h, corresponding to a turnover number (TON = [calcitriol]_final_ × [*Aae*UPO]^−1^) of 4000 of the enzyme. Increasing reaction time to 24 h resulted in a slightly decreased target calcitriol formation, which could be due to overoxidation reaction, leading to by-products. The *Aae*UPO showed strict regioselectivity and yielded only calcitriol, as judged from the chromatogram, which was carefully confirmed by comparison with the commercial standard compound (Figure 2B). We further analyzed calcitriol via LC–MS. As shown in Figure 2C, the ESI–MS spectrum exhibits a major peak at *m*/*z* 417.3346 [M + H]^+^, and the molecular mass error with calcitriol standard product is 0.2 ppm. The control reactions that either used thermally inactivated enzyme (boiled at 95 °C for 5 min) or used only H_2_O_2_ did not yield any product, suggesting a true enzymatic oxyfunctionalization reaction of C-H bonds at the C-25 position of alfacalcidol. As the peroxide concentration and enzyme concentration are crucial parameters in peroxygenase catalysis, we also systematically investigated their influence on calcitriol synthesis. An increase in the enzyme concentration to 2.5 μM resulted in an increase in product concentration (Figure 2D), which could be explained by the relatively slow activity of peroxygenase toward alfacalcidol oxyfunctionalization. Under the highest enzyme concentration, an increase in H_2_O_2_ feeding rate from 1.0 to 1.5 mM h^−1^ yielded almost complete conversion of the substrate (2.2 mM, Figure 2E), whereas it decreased upon a further increase in H_2_O_2_, which could be rationalized by the inactivation of the biocatalyst caused by the accumulation of H_2_O_2_. Substrate concentration also had an influence on product formation. With an increase in the substrate’s concentration from 2.5 to 10 mM, product formation decreased from 94% to 37% (Appendix A). Using the optimal condition, we also performed a reaction on a 200 mL scale, obtaining a 72% isolated yield (Appendix A).

Next, we sought to determine whether it is possible to synthesize calcitriol directly from vitamin D3, which is a desirable strategy due to the commercial availability and low cost of vitamin D3, compared with alfacalcidol (Figure 3). In this way, calcitriol (**4**) synthesis can be obtained via the double hydroxylation at C-1 and C-25 of vitamin D3 (**1**), leading to alfacalcidol (**2**) and 25-hydroxyvitamin D3 (**3**), respectively. It was found that there was no formation of **2** from the oxidation of **1**, as shown in Figure 3. Instead, **3** was obtained as the major product (74% product formation). The results agreed with the report by Gutiérrez et al. [30,31]. Interestingly, an amount of calcitriol (24% product formation) was also obtained, which was not observed in any early reports (Figure 3 and Figure 4). This indicates that it is possible to synthesize calcitriol directly from vitamin D3; however, to switch the product distribution from **3** to **4**, protein engineering of *Aae*UPO is necessary for future studies.

Meanwhile, it is worth noting that we also investigated the potential substrate and product prohibition of the *Aae*UPO. As shown in Figure 5A, incubation with vitamin D3, 25-hydroxyvitamin D3, and alfacalcidol showed no effect on the enzyme activity; only an inhibition effect of calcitriol on the enzyme was observed. This inhibition effect was further confirmed by increasing the calcitriol concentration incubated with *Aae*UPO (Figure 5B).

To further understand the performance of *Aae*UPO in the calcitriol synthesis, molecular docking was performed. The three compounds (**1**, **2**, and **3**) were docked to the active site of Pada-I, and the best-scored docking pose was acquired from each docking to predict their binding modes and also to explain the observed differences in activities (Figure 6). The docking results showed that with a binding energy of −8.37 kcal/mol, the alkyl chain of compound **1** oriented toward the Fe=O center, suggesting that *Aae*UPO is well capable of catalyzing the hydroxylation of the C-25 position. In the case of compound **2**, the best docking pose was also a catalytic conformation—namely, the alkyl chain end (C-25), which was oriented toward the Fe=O center. However, for compound **3**, the cyclohexane ring was far away from the Fe=O center, indicating that *Aae*UPO cannot efficiently catalyze the hydroxylation of **3** to the final product **4**. In another pose (Figure 6D), it was observed that the hydroxyl group at the C-3 position could form a hydrogen bond with the O atom of the Fe=O center, with a lower binding energy of −3.87 kcal/mol, which enhanced the possibility of the A ring oxidation of vitamin D3. This could be the reason that a small amount of **4** was observed in the reaction starting from the oxygenation of **1**.

## 4. Conclusions

In summary, we demonstrated that peroxygenase from *Agrocybe aegerita* can catalyze the selective C-25 hydroxylation of alfacalcidol yielding calcitriol, which represents a new enzymatic approach to synthesizing calcitriol. A maximum turnover number of 4000 of the enzyme was achieved, suggesting room for further improvement of the catalytic efficiency, compared with previous benzylic and other aliphatic C-H bonds oxidation reactions. It is also possible, in principle, to synthesize the calcitriol directly starting from vitamin D3. However, the low activity needs to be further addressed by protein engineering.

## Data Availability

Data is contained within the article and Appendix A.

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
