# Peer review of "Peroxygenase-Catalyzed Selective Synthesis of Calcitriol Starting from Alfacalcidol"

_antioxidants, 2022, doi:10.3390/antiox11061044_

Round 1
Reviewer 1 Report
The authors perform the monohydroxylation of alfacalcidol at position C25 by a known unspecific peroxygenase from Agrocybe aegerita (AaeUPO), yielding the target calcitriol. Additionally, the sequential dihydroxylation of the precursor vitamin D3, yielding calcitriol via the intermediate hydroxyvitamin D3, is suggested.
Supported by good English, the short length of the article, and the manageable number of results, the authors' claims are easy to follow. The authors apply a repertoire of well-established techniques related to AaeUPO including expression in yeast as well as the analysis of target vitamin D3-derived compounds.
However, the article misses (experimental) details that will ensure reproducibility, clarity (e.g., the use of the term conversion), and - as far as I am concerned - depth.
In Figure 2, depicting results from alfacalcidol to calcitriol transformations (performed in vitro with purified AaeUPO at 2.5 mM substrate load), lack error bars (B, D, and E). What do data points depict (e.g., mean values)? How many and what kind of replicates were performed? This information must be depicted graphically as error bars (e.g., standard deviation) and explained sufficiently in the figure legend. Although error bars are shown in Figure 5, information about representation/calculation are missing.
Since it is not clearly deducible from the main text and the graphic representations either: By conversion, do the authors mean substrate consumption or product formation (i.e., yield)?
The authors claim conversion of 76% and 98% for biotransformations starting from alfacalcidol (pathway I) and vitamin D3, respectively. Do these conversions reflect isolated yields of the corresponding products, calcitriol and a mixture of hydroxyvitamin D3 (74%)/calcitriol (24%), respectively. Can conversions be supported by isolated yields? Upscaling was performed at least for pathway I. (NMR analysis is acknowledged but corresponding data not presented.) It is OK to choose arbitrarily conditions (e.g., 2.5 mM substrate load). However, a couple of reaction parameters were tested but not the substrate load. The substrate load is also different from referenced studies employing, for example, 0.5 mM of vitamin D and anologs (Lucas et al., 2016). The same study already hydroxylates vitamin D3 at position C-25 yielding hydroxyvitamin D3 with the same peroxygenase. (Lucas et al. did not report the detection of the second hydroxylation yielding calcitriol by GC/MS).
Along these lines and regarding the two-step hydroxylation, I would not call 24% conversion a minor amount. - Again, I recommend the authors to use the terms substrate consumption and product formation for clarity. - What is the retention time of vitamin D3? A reference chromatogram is not provided; Figure 4 only shows the hydroxylated products from a reaction. Vitamin D3 is also missing in the Materials/Method section as reference compound as well, indicating it was not quantified in this study. Since the further hydroxylation of hydroxyvitamin D3 is not supported by docking, did the authors find other docking poses (or any other experiments/even speculations) explaining their results?
The authors suggest by docking and experimentally show that the hydroxylation of vitamin D3 does not yield alfacalcidol. Hence, I suggest to re-draw/adapt Scheme 1 for clarity.
HPLC/UV is reported in the Materials/Methods section but MS analysis is missing. This relates also to insufficient information in the legends of Figure 2C and Figure 4.
Although the two sequential hydroxylations of vitamin D3, leading to the value-added and therapeutically relevant calcitriol, by a single enzyme are intriguing, it is the only presented in a very preliminary way; this is also emphasized by the choice of the title.
Further, the AaeUPO enzyme has already been introduced for the first hydroxylation, yielding the intermediate hydroxyvitamin D3. The second (newly tested) substrate is the structurally-related alfacalcidol, yielding calcitriol as well.
The corresponding vitamin D2 analogs are accepted by AaeUPO (Lucas et al., 2016) but were not tested in this work. Although hydroxylated vitamin D3 analogs have been suggested to be therapeutically more active, vitamin D2 (analogs) can be derived from plants, which should not be disregarded in the light of growing dietary, social, and environmental concerns.
In summary, the presented work misses substantial improvement to existing protocols. Isolated yields from preparative biotransformations are missing for all biotransformations, for example. The substrate scope has been marginally expanded. Reaction engineering is preliminary and is claimed to be investigated by the authors in the future. Docking was performed and supports the experimental results only for the single hydroxylation steps. The authors are aware that protein engineering might increase the activity of AaeUPO towards the sequential hydroxylations, yielding calcitriol, but has not been performed yet and will be investigated in the future. Hence, I'd like to invite the authors to perform additional experiments for depth and to enhance their manuscript. Minimal statistical analysis for all biotransformations must be provided.
Minor comments
The experimental section misses details that would increase reproducibility, for example:
How long were pre-incubations with substrates before supplying H2O2. What was the total volume of reactions? How often was extracted with EtOAc and which volumes were used? After combination of the organic phases, what was the drying agent (Na2SO4?) and was EtOAc removed completely? - The latter is implied since in section 2.4., samples were dissolved in EtOH before HPLC measurment.
Indicate if LC refers to HPLC and which detector was used (UV or MS). MS analysis is missing in the Materials/Methods section. Please, add.
Use formulas (as in 2.3.,) for consistency including consistent formatting. Please, provide all values including constants (e.g., ε).
The "turnover number" was mentioned to support results. How was it calculated?
Line 108: 420 nm not "420nm" (Which microplate reader was used? Please, provide type and company for key resources including equipment and chemicals.)
Line 128: to make not "to made"; better: to perform standard calibration
134: to simplify not "to simply"
155: Please, indicate color (black) of the chromatogram of the reaction solution
156: Update figure legend (n = x, values represent etc.)
186f: Do the authors mean (substrate/product) inhibition? (not "prohibition")
190: Update figure legend (n = x, values represent etc.)
194: Consider moving detailed description of docking to the Material/Methods section.
211: Figure legend insufficient. Why certain residues are highlighted (e.g., S191, G241).
Author Response
Comments and Suggestions for Authors
The authors perform the monohydroxylation of alfacalcidol at position C25 by a known unspecific peroxygenase from Agrocybe aegerita (AaeUPO), yielding the target calcitriol. Additionally, the sequential dihydroxylation of the precursor vitamin D3, yielding calcitriol via the intermediate hydroxyvitamin D3, is suggested.
Supported by good English, the short length of the article, and the manageable number of results, the authors' claims are easy to follow. The authors apply a repertoire of well-established techniques related to AaeUPO including expression in yeast as well as the analysis of target vitamin D3-derived compounds.
However, the article misses (experimental) details that will ensure reproducibility, clarity (e.g., the use of the term conversion), and - as far as I am concerned - depth.
A: First of all, we would like to thank the reviewer for his/her rather positive and critical comments to help improve the quality of the manuscript. We have taken all comments seriously and revised manuscript by implementing extra experiments and rewriting.
In Figure 2, depicting results from alfacalcidol to calcitriol transformations (performed in vitro with purified AaeUPO at 2.5 mM substrate load), lack error bars (B, D, and E). What do data points depict (e.g., mean values)? How many and what kind of replicates were performed? This information must be depicted graphically as error bars (e.g., standard deviation) and explained sufficiently in the figure legend. Although error bars are shown in Figure 5, information about representation/calculation are missing.
A: We fully agree with the comments here and we are sorry for the floppy plotting and writing here. In revised version, these experiments have been performed in duplicates, the error bars have been added.
Since it is not clearly deducible from the main text and the graphic representations either: By conversion, do the authors mean substrate consumption or product formation (i.e., yield)?
A: We are sorry for the confusion here. As already present in the original manuscript (section 2.2 Enzymatic calcitriol synthesis from alfacalcidol), the conversion was calculated by [product]/([substrate]+ [product])Í100%. Therefore, it refers to the product formation. Following the reviewer’s other suggestion, we have modified the sentences accordingly by using product formation throughout the entire manuscript.
The authors claim conversion of 76% and 98% for biotransformations starting from alfacalcidol (pathway I) and vitamin D3, respectively. Do these conversions reflect isolated yields of the corresponding products, calcitriol and a mixture of hydroxyvitamin D3 (74%)/calcitriol (24%), respectively. Can conversions be supported by isolated yields? Upscaling was performed at least for pathway I. (NMR analysis is acknowledged but corresponding data not presented.)
A: We fully understood the concern from the reviewer. As already stated in the above-mentioned answers, the conversion was calculated by HPLC analysis, which is not related to isolated yield. We have now clarified more clearly in the manuscript. For the upscaling reaction, we have performed this reaction again and the isolation yield is now reported.
It is OK to choose arbitrarily conditions (e.g., 2.5 mM substrate load). However, a couple of reaction parameters were tested but not the substrate load. The substrate load is also different from referenced studies employing, for example, 0.5 mM of vitamin D and anologs (Lucas et al., 2016).
A: We thank the reviewer for this suggestion, we have now implemented the experiments varying substrate concentration. The results have now been included into the SI.
The same study already hydroxylates vitamin D3 at position C-25 yielding hydroxyvitamin D3 with the same peroxygenase. (Lucas et al. did not report the detection of the second hydroxylation yielding calcitriol by GC/MS).
A: We are certainly aware of the work by Lucas et al., 2016. It is true that they did not report the formation of calcitriol starting from vitamin D3. Our experience with the vitamin D compounds is that the analytics is very challenging, this is probably the reason that it was not observed in previous study.
Along these lines and regarding the two-step hydroxylation, I would not call 24% conversion a minor amount. - Again, I recommend the authors to use the terms substrate consumption and product formation for clarity.
A: We thank the reviewer for his/her suggestion. We have modified the sentences accordingly by using product formation throughout the entire manuscript.
- What is the retention time of vitamin D3? A reference chromatogram is not provided; Figure 4 only shows the hydroxylated products from a reaction. Vitamin D3 is also missing in the Materials/Method section as reference compound as well, indicating it was not quantified in this study.
A: We are sorry that the information was not presented enough. In the HPLC method we developed, the retention time of vitamin D3 was longer than 100 min, also the retention time of this compound was shifting from time to time. However, we did not come across this problem with three other compounds (i.e. alfacalcidol, 25-Hydroxyvitamin D3 and calcitriol). Thus, to get reliable results, all other three compounds were quantified by calibration curves. As such, the concentration and product formation starting either from alfacalcidol or vitamin D3 were reliable. We hope that the reviewer will understand this circumstance. Necessary information has been updated in the revised manuscript.
Since the further hydroxylation of hydroxyvitamin D3 is not supported by docking, did the authors find other docking poses (or any other experiments/even speculations) explaining their results?
A: We indeed observed another pose of 25-OH-D3 with binding energy of -3.87 kcal/mol in the docking study, which indicates the possibility of hydroxylation at C-1 position. This updated information has been added in the revised manuscript.
The authors suggest by docking and experimentally show that the hydroxylation of vitamin D3 does not yield alfacalcidol. Hence, I suggest to re-draw/adapt Scheme 1 for clarity.
A: The suggestions by the reviewer are highly appreciated! We have modified Scheme 1 for clarity.
HPLC/UV is reported in the Materials/Methods section but MS analysis is missing. This relates also to insufficient information in the legends of Figure 2C and Figure 4.
A: We are very sorry the missing information. The information has now been included.
Although the two sequential hydroxylations of vitamin D3, leading to the value-added and therapeutically relevant calcitriol, by a single enzyme are intriguing, it is the only presented in a very preliminary way; this is also emphasized by the choice of the title. Further, the AaeUPO enzyme has already been introduced for the first hydroxylation, yielding the intermediate hydroxyvitamin D3. The second (newly tested) substrate is the structurally-related alfacalcidol, yielding calcitriol as well.
The corresponding vitamin D2 analogs are accepted by AaeUPO (Lucas et al., 2016) but were not tested in this work. Although hydroxylated vitamin D3 analogs have been suggested to be therapeutically more active, vitamin D2 (analogs) can be derived from plants, which should not be disregarded in the light of growing dietary, social, and environmental concerns.
A: We fully agree with the comments and appreciate the expertise of the reviewer very much. Indeed, AaeUPO enzyme has already been introduced for the first hydroxylation of vitamin D3, and in general, peroxygenases have shown tremendous potential in organic synthesis. Vitamin D2 was studied by Lucas et al, in fact, we also investigated the C-25 hydroxylation of vitamin D2 and obtained the corresponding 25-hydroxylated vitamin D2. However, as already reflected in the title of the manuscript, the main focus of this manuscript is to provide a new synthetic pathway of calcitriol, which is typically very challenging in organic chemistry. In order not to dilute the focus of the manuscript we prefer not to include the study of vitamin D2. We hope that the reviewer will agree with us. In case not, we can include the relevant data in the second round of the revision.
In summary, the presented work misses substantial improvement to existing protocols. Isolated yields from preparative biotransformations are missing for all biotransformations, for example. The substrate scope has been marginally expanded. Reaction engineering is preliminary and is claimed to be investigated by the authors in the future. Docking was performed and supports the experimental results only for the single hydroxylation steps. The authors are aware that protein engineering might increase the activity of AaeUPO towards the sequential hydroxylations, yielding calcitriol, but has not been performed yet and will be investigated in the future. Hence, I'd like to invite the authors to perform additional experiments for depth and to enhance their manuscript. Minimal statistical analysis for all biotransformations must be provided.
A: Again, we thank the reviewer for his/her significant effort in evaluating our manuscript and helping improve the quality of the manuscript. Error bars based on duplicates/triplicates have been included. We agree that the present data are a bit preliminary, however, with the encouraging results from the viewpoint of a prominent compound (calcitriol) synthesis and new application of peroxygenase, we believe the manuscript will gain attention from the fields of biocatalysis, organic synthesis and enzymology. Based on the extra experiments implemented and the comments by the reviewer, we have substantially modified the manuscript.
Minor comments
The experimental section misses details that would increase reproducibility, for example:
How long were pre-incubations with substrates before supplying H2O2. What was the total volume of reactions? How often was extracted with EtOAc and which volumes were used? After combination of the organic phases, what was the drying agent (Na2SO4) and was EtOAc removed completely? - The latter is implied since in section 2.4., samples were dissolved in EtOH before HPLC measurment.
A: We are sorry for any confusion and missing information. Based on the reviewer’s comments, we have revised the experimental section carefully.
Indicate if LC refers to HPLC and which detector was used (UV or MS). MS analysis is missing in the Materials/Methods section. Please, add.
A: We are sorry for the missing information. We have now revised the experimental section carefully.
Use formulas (as in 2.3.,) for consistency including consistent formatting. Please, provide all values including constants (e.g., ε).
A: Corrected.
The "turnover number" was mentioned to support results. How was it calculated?
A: Information added.
Line 108: 420 nm not "420nm" (Which microplate reader was used? Please, provide type and company for key resources including equipment and chemicals.)
A: Information added.
Line 128: to make not "to made"; better: to perform standard calibration.134: to simplify not "to simply"
A: Corrected.
155: Please, indicate color (black) of the chromatogram of the reaction solution
A: Information added.
156: Update figure legend (n = x, values represent etc.)
A: Updated.
186f: Do the authors mean (substrate/product) inhibition? (not "prohibition")
A: We are sorry for the unproper use of prohibition. Yes, inhibition was used in the corrected version.
190: Update figure legend (n = x, values represent etc.)
A: Updated.
194: Consider moving detailed description of docking to the Material/Methods section.
A: Correction was made according to the reviewer’s suggestion.
211: Figure legend insufficient. Why certain residues are highlighted (e.g., S191, G241).
A: the legend has been corrected.

Reviewer 2 Report
The authors in this article wish to enzymatically synthesize calcitriol, an active analog of vitamin D3, which has excellent physiological activity to ensure healthy immune function. This would avoid total syntheses that are sometimes long and tedious. It is a very good idea however although conversions are given it is not shown the possible preparative separation of the compounds. Indeed, only analytical data are present. However, to use this enzymatic pathway it would be necessary to prove that it is possible for synthetic purposes.
- It would therefore be necessary to show a preparative HPLC following this hydroxylation of the C-H bond with AaeUPO and to give the NMRs spectrum of the products obtained in the experimental part as well as in one supporting information.
- The small peak at the base of the calcitriol and alphacalcidol signal in Figure 2 needs to be explained.
- Diagram 1 does not agree with figure 3
A proofreading of the article must be done because there are some imperfections such as for example caltriol instead of Calcitriol line 166, 171, 173, 194, 216, 219....
In this case only, the article could possibly be accepted.
Author Response
Comments and Suggestions for Authors
The authors in this article wish to enzymatically synthesize calcitriol, an active analog of vitamin D3, which has excellent physiological activity to ensure healthy immune function. This would avoid total syntheses that are sometimes long and tedious. It is a very good idea however although conversions are given it is not shown the possible preparative separation of the compounds. Indeed, only analytical data are present. However, to use this enzymatic pathway it would be necessary to prove that it is possible for synthetic purposes. It would therefore be necessary to show a preparative HPLC following this hydroxylation of the C-H bond with AaeUPO and to give the NMRs spectrum of the products obtained in the experimental part as well as in one supporting information.
A: We thank the reviewer for spending effort in evaluating our manuscript. And also we fully agree with the suggestions that a preparative synthesis is needed to show the synthetic potential of the enzymatic process. The experiment has been implemented again and the analytical data are now included in Supporting Information.
The small peak at the base of the calcitriol and alphacalcidol signal in Figure 2 needs to be explained.
A: as shown in Figure 2B, calcitriol and alphacalcidol signal were from commercial standard compounds, therefore we believe it belongs to some impurities of the compounds. We also checked the reaction mixture at 0 h, these small peaks were also appeared.
Diagram 1 does not agree with figure 3
A: based on the comments from both reviewers 1 and 2, we have now changed the Scheme 1 accordingly.
A proofreading of the article must be done because there are some imperfections such as for example caltriol instead of Calcitriol line 166, 171, 173, 194, 216, 219.... In this case only, the article could possibly be accepted.
A: We are very sorry for the floppy writing. We have now checked the manuscript carefully and corrected possible errors.

Reviewer 3 Report
1) Page 1, Line 14: ...multiple steps ...
2) Page 1, Line 18: This formulation should be improved.
3) Page 1, Line 29: After the general comments on the vitamin D group, the connection to the specific group member vitamin D3 should be introduced.
4) Page 2, Line 54: ... for a specific electron transport chain ...
5) Page 2, Line 59: This formulation is not entirely correct, as the pilot-scale production mentioned refers to the pilot-scale production of the enzyme yielding practical amounts of these promising catalysts.
6) Page 2, Line 71: ... for a straightforward synthetic route ...
7) Page 3, Line 89: In the definition of the unit of enzyme activity the pH and the temperature are missing.
8) Page 3, Line 116: The actual number should be given here which has been used for ε as the molar absorbance value under the assay conditions.
9) Page 3, Line 121: This formulation should be improved and the supplier name of the C18-HPLC column used should be specified.
10) Page 4, Line 130: ... set of experiments ... ... the PaDa-I variant ....
11) Page 4, Line 134: ... simplify ...
12) Page 4, Line 137: The turnover number of 3980 should be explained in more detail.
13) Page 4, Line 149: ... rate of addition ...
14) Page 4, Line 150: ... decreased significantly upon a further increase ...
15) Page 4, Line 152: ... at 250 mL scale. What was the rate of hydrogen peroxide addition?
16) Page 5, Line 162: Figure 2E and the legend do not match and the x-axis should correctly describe the rate of hydrogen peroxide addition.
17) Page 5, Line 166: ... calcitriol ...
18) Page 5, Line 171: ... calcitriol ...
19) Page 5, Line 174: .. in a future study.
20) Page 5, Line 186: What do the authors mean with prohibition?
21) Page 5, Line 187: The details for the experiments shown in figure 5 are missing. Has the influence of increasing concentrations of calcitriol on the enzyme activity been measured? It would be useful to include this as a figure 5B.
22) Page 6, Line 194: ... calcitriol ...
23) Page 7, Line 216: see previous comments
24) Page 7, Line 219: ... calcitriol ...
25) Page 7, Line 221: To improve the substrate/product loading is not a conclusion, it should be described in an additional figure 5B. The statement that efforts via reaction engineering are currently ongoing is not entirely adequate as a conclusion.
Author Response
Comments and Suggestions for Authors
1) Page 1, Line 14: ...multiple steps ...
A: We are very sorry for the floppy writing. We have now checked the manuscript carefully and corrected the errors.
2) Page 1, Line 18: This formulation should be improved.
A: Corrected as suggested.
3) Page 1, Line 29: After the general comments on the vitamin D group, the connection to the specific group member vitamin D3 should be introduced.
A: We appreciate this suggestion very much. The manuscript has been revised accordingly.
4) Page 2, Line 54: ... for a specific electron transport chain ...
A: Corrected.
5) Page 2, Line 59: This formulation is not entirely correct, as the pilot-scale production mentioned refers to the pilot-scale production of the enzyme yielding practical amounts of these promising catalysts.
A: We are sorry for this confusion. The text has been corrected according to the comments.
6) Page 2, Line 71: ... for a straightforward synthetic route ...
A: Corrected.
7) Page 3, Line 89: In the definition of the unit of enzyme activity the pH and the temperature are missing.
A: Missing information added.
8) Page 3, Line 116: The actual number should be given here which has been used for ε as the molar absorbance value under the assay conditions.
A: Missing information added.
9) Page 3, Line 121: This formulation should be improved and the supplier name of the C18-HPLC column used should be specified.
A: Missing information added.
10) Page 4, Line 130: ... set of experiments ... ... the PaDa-I variant ....
A: Corrected.
11) Page 4, Line 134: ... simplify ...
A: Corrected.
12) Page 4, Line 137: The turnover number of 3980 should be explained in more detail.
A: The definition and calculation were added.
13) Page 4, Line 149: ... rate of addition ...
A: Corrected.
14) Page 4, Line 150: ... decreased significantly upon a further increase ...
A: Corrected.
15) Page 4, Line 152: ... at 250 mL scale. What was the rate of hydrogen peroxide addition?
A: The semi-preparative scale synthesis has been implemented again and the updated results have been included in the revised manuscript and SI.
16 Page 5, Line 162: Figure 2E and the legend do not match and the x-axis should correctly describe the rate of hydrogen peroxide addition.
A: We have corrected the legend of Figure 2E
17) Page 5, Line 166: ... calcitriol ...
A: Corrected.
18) Page 5, Line 171: ... calcitriol ...
A: Corrected.
19) Page 5, Line 174: .. in a future study.
A: Corrected.
20) Page 5, Line 186: What do the authors mean with prohibition?
A: We are sorry for the misunderstanding. A more appropriate word “inhibition” was used.
21) Page 5, Line 187: The details for the experiments shown in figure 5 are missing. Has the influence of increasing concentrations of calcitriol on the enzyme activity been measured? It would be useful to include this as a figure 5B.
A: We thank the reviewer for his/her help comments, in the revised version the information has been added.
22) Page 6, Line 194: ... calcitriol ...
A: Corrected.
23) Page 7, Line 216: see previous comments
A: Corrected.
24) Page 7, Line 219: ... calcitriol ...
A: Corrected.
25) Page 7, Line 221: To improve the substrate/product loading is not a conclusion, it should be described in an additional figure 5B. The statement that efforts via reaction engineering are currently ongoing is not entirely adequate as a conclusion.
A: We are grateful for the comments herein. We have now revised the conclusion part.

Round 2
Reviewer 1 Report
Li and co-workers re-submitted their manuscript "Perogxygenase-Catalyzed Selective Synthesis of Calcitriol Starting from Alfacalcidiol" and have addressed most of my concerns of the first evaluation. Hence, I recommend publication of the article with "Antioxidants" after minor revision, especially checking for: missing spaces, spelling, formatting etc. The authors are also asked to add a short paragraph regarding the HPLC analysis/challenges of vitamine D3.
Author Response
Q: Li and co-workers re-submitted their manuscript "Perogxygenase-Catalyzed Selective Synthesis of Calcitriol Starting from Alfacalcidiol" and have addressed most of my concerns of the first evaluation. Hence, I recommend publication of the article with "Antioxidants" after minor revision, especially checking for: missing spaces, spelling, formatting etc. The authors are also asked to add a short paragraph regarding the HPLC analysis/challenges of vitamine D3.
A: We thank the reviewer for his/her effort in helping improve our manuscript. The manuscript has been checked carefully regarding to the typo and format errors.
A brief description of the relevant HPLC analysis/challenges for vitamin D3 has also been included in the section 2.5 ‘HPLC analysis’.

Reviewer 2 Report
This article can be accepted in present form
Author Response
Q: This article can be accepted in present form.
A: Thank you!

Reviewer 3 Report
The reviewer comments have all been addressed. The revised manuscript has been improved and can be accepted after some small text corrections (see below).
Comments to Revised Manuscript Antioxidants2022:
1) Page 6, Line 201: ... oxyfunctionalization ...
2) Page 8, Line 245: Reaction conditions ...
3) Page 8, Line 246: Reaction conditions ...
Author Response
Q: The reviewer comments have all been addressed. The revised manuscript has been improved and can be accepted after some small text corrections (see below).
Comments to Revised Manuscript Antioxidants2022:
1) Page 6, Line 201: ... oxyfunctionalization ...
2) Page 8, Line 245: Reaction conditions ...
3) Page 8, Line 246: Reaction conditions ...
A: We thank the reviewer for his/her evaluation and the manuscript has been corrected accordingly based on the comments.
